# A Co-Printed Nanoslit Surface Plasmon Resonance Structure in Microfluidic Device for LMP-1 Detection

**DOI:** 10.3390/bios12080653

**Published:** 2022-08-17

**Authors:** Shu-Cheng Lo, Shao-Sian Li, Wen-Fai Yang, Kuang-Chong Wu, Pei-Kuen Wei, Horn-Jiunn Sheen, Yu-Jui Fan

**Affiliations:** 1Institute of Applied Mechanics, National Taiwan University, No. 1, Section 4, Roosevelt Road, Taipei 10614, Taiwan; 2Research Center for Applied Sciences, Academia Sinica, 128 Academia Road, Section 2, Nankang, Taipei 11529, Taiwan; 3Department of Materials and Mineral Resources Engineering, National Taipei University of Technology, 1, Section 3, Zhongxiao E. Rd., Taipei 10608, Taiwan; 4Graduate Institute of Nanomedicine and Medical Engineering, International PhD Program in Biomedical Engineering, School of Biomedical Engineering, International PhD Program in Biomedical Engineering, International PhD Program for Cell Therapy and Regeneration Medicine Taipei Medical University, 250 Wuxing St., Taipei 11031, Taiwan

**Keywords:** hot embossing, microfluidics, localized surface plasmon resonance (SPR), immunoassay

## Abstract

This paper reports a novel micro/nanostructure co-hot embossing technique. Gold-capped nanostructures were used as localized surface plasmon resonance (SPR) sensors and were integrated into a microfluidic channel. The advantage of the co-hot embossing technique is that the SPR sensors do not need to be aligned with the microfluidic channel while bonding to it. The integrated SPR sensor and microfluidic channel were first characterized, and the sensitivity of the SPR sensor to the refractive index was found using different concentrations of glycerol solutions. The SPR sensor was also used to quantify latent membrane protein (LMP-1) when modifying anti-LMP-1 at the surface of the SPR sensor. Different concentrations of LMP-1 samples were used to build a calibration curve.

## 1. Introduction

Latent membrane protein 1 (LMP1) is an important biomarker of the Epstein–Barr virus (EBV), which is related to nasopharyngeal carcinoma (NPC) when transfected. NPC is a rare tumor of head and neck, originating in the nasopharynx. NPC is a common occurrence in the regions of East Asia and Africa. With this in mind, it is recommended to frequently check for EBV when NPC-suspected persons are in these regions [1,2]. To reliably detect LMP1 in patients leaving a hospital, portable biosensors are necessary. 

Many types of biosensors (e.g., electrochemical [3,4,5,6], immunobeads motion [7,8,9], mechanical, optical types [10,11,12,13,14,15,16,17,18,19,20] etc.) have been developed and show the potential for becoming portable. From these efforts, optical type biosensors, such as localized surface plasmon resonance (LSPR) and surface-enhanced Raman spectroscopy (SERS) have been demonstrated to have high sensitivity and high accuracy. The phenomenon of LSPR is that conductive electrons coherently oscillate to incident light. When the light propagates into the metallic nanostructure, the electric field near the surface of the metal is enhanced and has a maximum resonant frequency that can be determined by the size and shape of the metallic structure, the type of metal, and the environmental dielectric constants [21,22,23]. Suitable metals, such as silver, Pd, and gold, are used to produce NPs with different shapes for localized SPR generation, due to the exhibited spectrum in the visible range [24,25,26,27,28]. For biological applications, gold is normally considered to be ideal due to its inert nature and biocompatibility [21]. Moreover, for biosensing, the high association between thiol groups and gold makes biomolecular immobilization easier for metal surfaces.

Recently, a nanostructure on a polymer substrate produced by nanoimprinting and then depositing metal on the nanostructure was developed for biosensing [29,30,31]. The sensing chips can be quantified by analyzing the resonant spectrum and images [32]. Multiplex detection was also exhibited [33,34]. Biosensing applications were presented to quantify DNA/RNA/micro (mi)RNA [35,36,37,38], proteins [39], autoantibodies [40], cells [41], and NPs in living cells [42]. To reduce interference from environmental elements, the integration of biosensors into the microfluidic channel is a good strategy to isolate sensing environments [43,44,45]. Other advantages associated with using microfluidics are that these systems use less sample material, are easy to integrate with microfluidics, and can utilize multiplex detection. However, microfluidic channels are not easy to align and bond to nanostructural SPR biosensors due to the microscale size and structural configurations. A technician with special expertise is required to handle the process.

In this study, we developed a co-hot embossing method to simultaneously print nanostructural biosensors and microstructural channels on cyclic olefin polymer (COP) film without alignment. After depositing gold on the nanostructure and bonding polydimethylsiloxane (PDMS) to the microchannel, a biosensor integrated with the microfluidic channel was obtained. Hot embossing molding method showed advantages in that the devices can be quickly and massively produced on polymer substrates. The sensitivity of the developed localized SPR sensor embedded in the microfluidic channel was investigated using glycerol solutions with different weight percentages. To demonstrate the biomolecular sensing by the localized SPR sensing chip integrated in a microfluidic chip, the antibodies were modified on the sensing region to quantify LMP-1 concentration based on resonant wavelength shift of the localized SPR.

## 2. Materials and Methods

### 2.1. Micro/Nanostructural Mold Preparation

The micro/nanostructural mold was made by two-time lithography including e-beam lithography and traditional photo-lithography. Firstly, the nanoslit structure was patterned by e-beam lithography. The period of the nanoslit was 500 nm, the crest was 100 nm in width, the depth was 100 nm, and the total size was 150 μm × 150 μm on a silicon wafer. The microfluidic mold was made on the wafer and aligned with a grating structure. A depth of the microfluidic mold of 10 µm was obtained.

### 2.2. Hot Embossing Process

The fabricated micro/nanostructure was used as a mold for hot embossing. The COP film (ZF16-188, ZeonorFilm, Tokyo, Japan) with a thickness of 188 μm was cut to 3 × 4 cm, sandwiched between the mold and a flat glass wafer, and put into the hot embossing machine. Initially, the top/bottom plate was set to heat up to 180/140 °C, and a pressure of 0.20 MPa was provided for 90 s. After releasing the pressure, the top/bottom plate was cooled to 100/80 °C, and then the COP film was peeled off the mold.

### 2.3. Optical System and Sensing Mechanism

To measure the resonant wavelength of the localized SPR sensing chip, an optical setup was developed as shown in Figure 1a. A halogen lamp with 100-W broadband white light was used to illuminate the SPR chip. The white light was first polarized in the transverse magnetic (TM) direction and propagated to a 10× objective lens. The TM wave light was focused on the SPR region in the microfluidic chip and collected by another 10× objective lens on the opposite side. The collected light was guided into a spectrometer through an optical fiber. The resonant spectrum was recorded by a computer. Figure 1b–e shows the detection mechanism. 

## 3. Results and Discussion

The microstructure and nanostructure printing on the COP thin film were characterized by a surface profiler. Results in Figure 2 show that the microfluidic channel height was 8.7 µm, and the nanoslit was 100 nm in height.

After characterizing the structure, a thin gold film was deposited on the nanostructure to become the localized SPR sensor. First, 3M tape (8003p, 3M, Saint Paul, MI, USA) with a proper opening, illuminated by a CO_2_ laser, was prepared as the shadow mask. The shadow tape was aligned and stuck onto the COP substrate for depositing gold only on the nanostructure area as shown in Figure 3a. DC sputter with power of 0.06 kW was used to deposit the gold for different deposition times, and transmission rates of visible light in DI water were measured as shown in Figure 3b. At a deposition time of 60 s, the transmission rate began to dramatically decrease at wavelengths of 550~700 nm. As a result, an optimal deposition time of 50 s was chosen for further experiments. After removing the tape, the microstructure and nanostructure were captured using a regular camera and scanning electron microscope (SEM), respectively, as shown in Figure 3c,d, respectively. The period of the nanoslit was 500 nm, and the crest was 100 nm in width.

After the gold deposition on the nanostructure, the COP film was bound to a piece of cured PDMS to become the microfluidic channel. The COP film was first treated with oxygen plasma, and then bathed in a (3-aminopropyl) tiethoxysilane (APTES)/DI water solution at 1% *v*/*v* for 20 min. After rinsing and drying, the COP film was ready for bonding. Holes were punched in the cured PDMS to connect tubes for inlets and outlets, and then it was treated with oxygen plasma. Subsequently, the PDMS and COP film were hard contacted and left on a 90 °C hotplate for 30 min for bonding. Images of the microfluidic channel are shown in Figure 3e,f. PDMS is not suitable to make nanoslits because due to the high elasticity of PDMS, after depositing metal, the metal film will have a lot of cracks.

The localized SPR sensor was calibrated with different refractive index solutions, which were glycerol/water solutions with *v*/*v* percentages of 0%, 2.5%, 5%, 7.5%, 10%, 12.5%, 15%, 17.5%, and 20%. The normalized resonant wavelengths of these glycerol solutions in the localized SPR sensing area are plotted in Figure 4a. Results indicated that with higher concentrations of the glycerol solution, the transmission spectrum of the SPR sensor showed more red-shifting in the TM direction because the refractive index was higher. The experimental peak values of these resonant wavelengths versus refractive indexes are plotted in Figure 4b. The peak values were linearly regressed, and a wavelength sensitivity of 423 nm/RIU was obtained. We also plotted the theoretical peak values in Figure 4b, and they show certain shifts compared to the experimental results. This is because the Fano-type resonant peak is an asymmetrical spectrum, and the theoretical peak value is an average value of peak and dip values. The experimental values only considered the peak values of the resonant spectrum.

To demonstrate immunosensing by the developed SPR chips in the microfluidic channel, 1 mg/mL anti-LMP-1, and LMP-1 samples at concentration of 3 × 10^1^, 3 × 10^2^, 3 × 10^3^, 3 × 10^4^, and 3 × 10^5^ ng/mL in a phosphate-buffered saline (PBS) solution were prepared. First, the microfluidic channel was filled with the 1 × PBS solution, and the resonant spectrum was measured and recorded. Subsequently, the 1 mg/mL anti-LMP-1 in PBS solution was allowed to flow into the microfluidic channel to modify anti-LMP-1 to the sensing region of the SPR chip. After incubating the anti-LMP-1 solution for 2 h, the resonant spectrum of the SPR chip was recorded, and the peak value was observed to have a 1.2-nm red-shift, compared to that with only the PBS solution. Different concentrations of LMP-1 samples were then continuously allowed to flow into the microfluidic channel at a flow rate of 10 µL/h for 10 min to dynamically conjugate the LMP-1 onto the anti-LMP-1 in the sensing region. The resonant spectra of these samples are plotted in Figure 5a.

After repeating the experiment three times, the average red-shift of each sample versus the base 10 logarithm of each concentration of LMP-1 solution was plotted in Figure 5b. The red-shift was linearly proportional to the logarithmic concentration, and an R-squared value of 0.98 was found. The results indicated that the developed biosensing chip can be used to quantify biomolecular concentration.

## 4. Conclusions

In this study, the designed microfluidic channel and nanoslit structure, microfabricated on a silicon substrate, were used as the mold for hot embossing. The micro- and nanostructure could be simultaneously transferred to COP film without alignment. After depositing gold onto the nanoslit and bonding with cured PDMS, the localized SPR sensor in a microfluidic channel was obtained. An optical system with a spectrometer was developed to collect the resonant spectra of localized SPR signals in the TM direction. The peak value of the resonant spectrum was analyzed for quantification of SPR sensing. The sensitivity of the SPR sensor was characterized using different concentrations of glycerol solutions. An immunoassay of the SPR sensor was also quantified through LMP-1 and anti-LMP-1 interactions. A calibration curve was also obtained by measuring different concentrations of LMP-1. It was demonstrated that the developed co-hot embossing technique can be used to fabricate a localized SPR sensor and microfluidic channel by one-time printing. It was also proven that the SPR sensor integrated with the microfluidic channel can quantify a label-free immunoassay.

## Figures and Tables

**Figure 1 biosensors-12-00653-f001:**
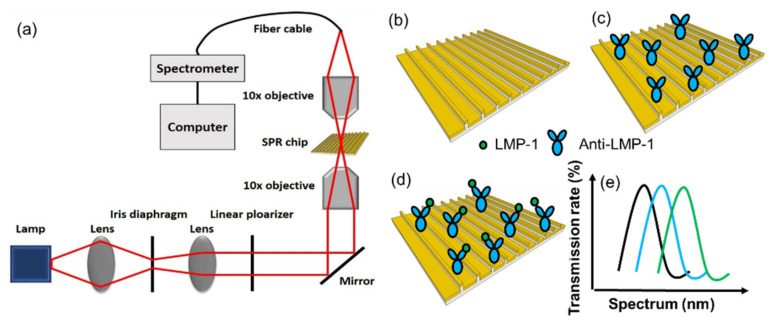
(**a**) Optical setup and (**b**–**e**) the detection mechanism.

**Figure 2 biosensors-12-00653-f002:**
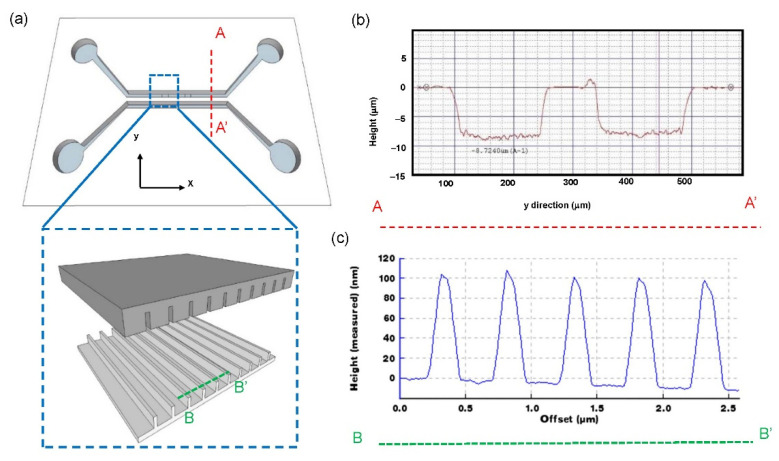
Surface profile of the microstructure and nanostructure. (**a**) Schematics of the microfluidic devices. The profiles of (**b**) microstructure and (**c**) nanostructure were measured.

**Figure 3 biosensors-12-00653-f003:**
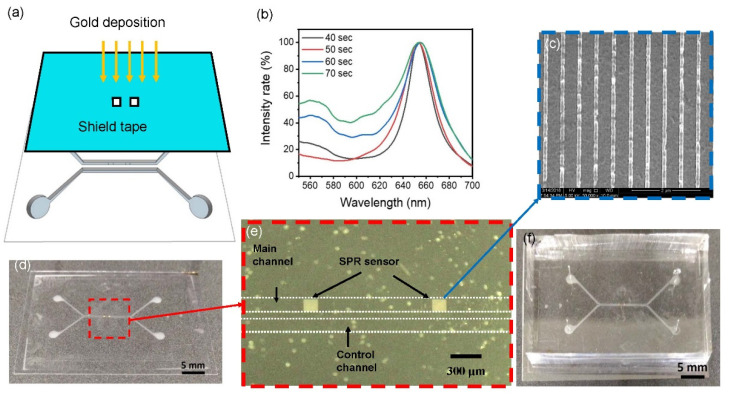
(**a**) Covering the opening with shadow tape and depositing gold. (**b**) The thickness of the gold deposition was related to the resonant spectrum. A thicker gold film possesses lower light transmission. (**c**–**f**) Macroscopic to microscopic views of the microstructure and nanostructure.

**Figure 4 biosensors-12-00653-f004:**
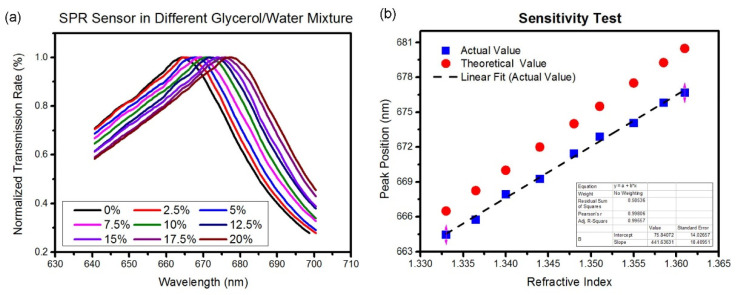
Sensitivity test. (**a**) Resonant spectra of the nanoslit SPR chip immersed in glycerol/water solutions with different refractive indices. (**b**) Curve fitting of the wavelengths of the resonant peaks.

**Figure 5 biosensors-12-00653-f005:**
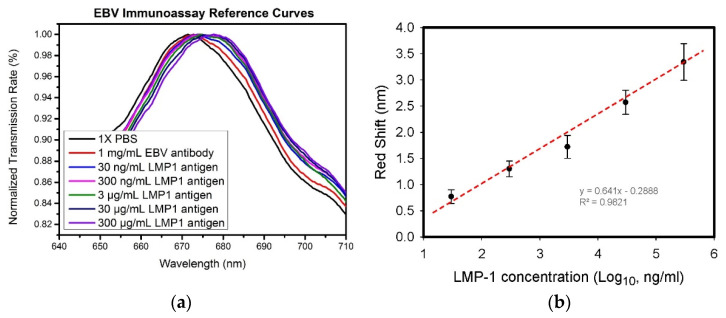
(**a**) Observation of resonant spectrum shifting when antigen and antibody were sequentially conjugated onto the nanoslit surface. (**b**) Analytical results of resonant spectra when the sensing chip was immersed in various concentrations of sample solutions. (Black dots and lines are averaged peak values and standard deviations of resonances when measuring various concentrations of LMP 1 antigen in three times, and red dot line is linear regression results of these averaged peak values).

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
