# Peer review of "A Co-Printed Nanoslit Surface Plasmon Resonance Structure in Microfluidic Device for LMP-1 Detection"

_biosensors, 2022, doi:10.3390/bios12080653_

Round 1
Reviewer 1 Report
The manuscript describes the fabrication and optical and biological validation of a microfluidic structure embedding an LSPR-based sensor used in combination with an external optical setup. The fabrication process is such that the nanostructure is embedded and aligned within the microchannel. The manuscript is concise enough, well written and adequately illustrated.
We note that in the fabrication process, once the Si mold is ready, the effect of embedding the LSPR structure in the microchannel is obtained for any way in which the molding is done, that is, it does not seem particularly dependent on hot co-embossing. For instance, once could mold PDMS onto the Si mold, and then finalize the structure by bonding the PDMS to glass. So we miss the added advantage of embossing, or don't find it emphasized enough.
Would the Au-coated LSPR structure work also in other substrate materials?
Figure 2a is confusing to us, as it does not seem to represent the mold itself but its negative already (the structure seems etched in, not in relief as in Figure 1i).
The link between the first and second paragraph of the Introduction is not consequential, the two sections seem simply juxtaposed.
Figure 4b is of low quality and hard to read.
It is not known from the text how the theoretical peak values in Figure 6b were calculated.
Author Response
Thanks for the valuable comments. We have uploaded the revised manuscript and answered the questions one by one raised by reviewers. Please send the outcomes of the review to me at your earliest convenience. Thank you and best regards.

Reviewer 2 Report
The authors patterned the nano-slits to excite SPR effect to determine the concentration change in a dual-channel microfluidic chip. The fabrication process was detailed explained and the measurements results for RI and LMP-1 were illustrated. I recommend a reject or major revision for its possible publication in Biosensors or other journals. In my opinion, there is no new finding or ideas in this paper. To the most important thing, the authors failed to reveal the detailed results or discuss the impact factors. Some comments can be found below to improve it.
1. Please note that the capitalization of the first letter of the word should be standardized, for example: “Mechanical” in line 40 should be “mechanical”; the unit of RI sensitivity should be nm/RIU instead of “nm/refractive index” in line 171;
2. Introduction should be re-written in a logic way. Here are some problems: line 49-50: the authors focused the SPR materials on silver and gold, but why Pd was inserted among the sentences? As well as for aluminum in line 55; the authors pay much more attentions on “Fano resonance”, however, in this work, only the most common SPR spectra were observed, it is meaningless to review or discuss the unrelated works here;
3. The fabrication process was detailed explained in the manuscript. I suggest the authors plot only one figure to replace figures 1-3, because some fabrication methods or process is well known for most of readers.
4. In figure 3, the authors illustrate the optical system and the detection mechanism. For the optical system, the TM polarized light was shined on the surface of nano-slits to excite the SPR effect, what about impact from the relative angles for nano-slits and light beam, the relative position of the focused light points near two microfluid channels; For the detection mechanism, why the BSA model was used instead of LMP1?
5. In figure 4, why the intensity for black line (40 sec) is higher than 1.0 on its left side?
6. Some other details should be discussed: for example: what are the different roles of two channels? whether the experimental results were obtained based on analyzing the spectra difference of the two channels? How about the impact of temperature? How about the stability and repeatability of the proposed sensor? etc.
Author Response

(The authors gave the same response as above.)

Round 2
Reviewer 2 Report
None
Author Response
Thanks for reviewing.